# ERK1/2-Dependent Phosphorylation of GABA_B1_(S867/T872), Controlled by CaMKIIβ, Is Required for GABA_B_ Receptor Degradation under Physiological and Pathological Conditions

**DOI:** 10.3390/ijms241713436

**Published:** 2023-08-30

**Authors:** Musadiq A. Bhat, Thomas Grampp, Dietmar Benke

**Affiliations:** 1Institute of Pharmacology and Toxicology, University of Zurich, 8057 Zurich, Switzerland; musadiq.bhat@pharma.uzh.ch (M.A.B.); tgr@pharma.uzh.ch (T.G.); 2Neuroscience Center Zurich, University of Zurich and ETH Zurich, 8057 Zurich, Switzerland

**Keywords:** GABA_B_ receptors, ERK1/2, CaMKII, phosphorylation, degradation, cerebral ischemia

## Abstract

GABA_B_ receptor-mediated inhibition is indispensable for maintaining a healthy neuronal excitation/inhibition balance. Many neurological diseases are associated with a disturbed excitation/inhibition balance and downregulation of GABA_B_ receptors due to enhanced sorting of the receptors to lysosomal degradation. A key event triggering the downregulation of the receptors is the phosphorylation of S867 in the GABA_B1_ subunit mediated by CaMKIIβ. Interestingly, close to S867 in GABA_B1_ exists another phosphorylation site, T872. Therefore, the question arose as to whether phosphorylation of T872 is involved in downregulating the receptors and whether phosphorylation of this site is also mediated by CaMKIIβ or by another protein kinase. Here, we show that mutational inactivation of T872 in GABA_B1_ prevented the degradation of the receptors in cultured neurons. We found that, in addition to CaMKIIβ, also ERK1/2 is involved in the degradation pathway of GABA_B_ receptors under physiological and ischemic conditions. In contrast to our previous view, CaMKIIβ does not appear to directly phosphorylate S867. Instead, the data support a mechanism in which CaMKIIβ activates ERK1/2, which then phosphorylates S867 and T872 in GABA_B1_. Blocking ERK activity after subjecting neurons to ischemic stress completely restored downregulated GABA_B_ receptor expression to normal levels. Thus, preventing ERK1/2-mediated phosphorylation of S867/T872 in GABA_B1_ is an opportunity to inhibit the pathological downregulation of the receptors after ischemic stress and is expected to restore a healthy neuronal excitation/inhibition balance.

## 1. Introduction

The balance between excitation and inhibition is a key principle for stable neuronal network activity and proper brain function. Its imbalance is considered the basis of pathophysiological mechanisms in many neurological disorders, where sustained neuronal overexcitation and diminished inhibition often result in neuronal death (excitotoxicity). Under physiological conditions, neuronal excitation is controlled, among other factors, by GABA_B_ receptors [1]. They are heterodimeric G-protein coupled receptors (GPCR) consisting of two subunits, GABA_B1_ (present in two variants: GABA_B1a_ and GABA_B1b_) and GABA_B2_ [2,3,4]. GABA_B_ receptors are expressed in the majority of neurons at pre- and postsynaptic locations [5]. The binding of the neurotransmitter GABA to the receptor activates G_i/o_ proteins and induces neuronal inhibition by modulating the activity of several effector systems. At postsynaptic sites, the notable effect of GABA_B_ receptors is the activation of G protein-coupled inwardly rectifying potassium (GIRK or Kir3) channels, which inhibits the generation of action potentials by hyperpolarizing the neuronal membrane [6,7]. At presynaptic sites, GABA_B_ receptors inhibit voltage-gated Ca^2+^ channels, resulting in reduced neurotransmitter release [8,9,10,11].

In neurological diseases associated with a disturbed excitation/inhibition balance, such as anxiety, addiction, depression, neurodegenerative diseases, and cerebral ischemia, GABA_B_ receptors are downregulated and thus are no longer able to counteract pathologically increased neuronal activity [12,13,14,15,16,17,18]. A major insight into a mechanism downregulating GABA_B_ receptors came from studies on neuronal excitotoxicity and cerebral ischemia, which are extreme examples of a disturbed excitation/inhibition balance. Under ischemic/excitotoxic conditions, GABA_B_ receptors are phosphorylated at serine 867 (S867) of GABA_B1_ by the calcium/calmodulin-dependent protein kinase II type β (CaMKIIβ) [19,20]. This phosphorylation event serves as a signal for K63-linked ubiquitination of the receptor by the E3 ubiquitin ligase Mind-bomb 2, which is required for targeting the receptors to lysosomes for degradation [20,21] via the endosomal sorting complexes required for transport (ESCRT) machinery [22].

A phospho-proteomics study revealed that GABA_B1_ can also be phosphorylated at threonine 872 (T872) [23], located in close proximity to S867 phosphorylated by CaMKIIβ. Thus, the question arose as to whether phosphorylation of T872 also contributes to targeting GABA_B_ receptors to lysosomal degradation and, if so, whether phosphorylation of S872 is mediated by CaMKIIβ or another protein kinase. Potential candidates for additional protein kinases involved in this mechanism are extracellular signal-regulated protein kinases ERK1 and ERK2 (alternative names MAP1 and MAP2) based on the observation that CaMKII can activate ERK1/2 [24,25,26,27,28,29]. In addition, like CaMKII, the ERK1/2 signaling pathway also plays a crucial role in cerebral ischemic-induced neuronal death [30]. Therefore, the aim of the present study was to analyze whether CaMKIIβ and/or ERK1/2 are phosphorylating T872 in GABA_B1_ and unravel the involvement of T872 phosphorylation in downregulating GABA_B_ receptor expression under physiological and ischemic conditions.

## 2. Results

### 2.1. GABA_B_ Receptor Expression Is Regulated by ERK1/2

Phosphorylation of GABA_B_ receptors mediated by CaMKIIβ triggers their lysosomal degradation [19,20]. To test whether ERK1/2 might contribute to this mechanism, we transfected HEK-293 cells with GABA_B1_ and GABA_B2_ alone or in addition to either ERK1 or ERK2. Two days after transfection, total and cell surface expression of GABA_B_ receptors were analyzed by immunofluorescence staining. The over-expression of ERK1 or ERK2 downregulated the total expression of GABA_B1_ (Figure 1A) and GABA_B2_ (Figure 1B) subunits and cell surface expression of GABA_B_ receptors as shown for GABA_B2_ (Figure 1C). These results suggest that ERK1/2 might be involved in triggering the degradation of GABA_B_ receptors.

To confirm the effect of ERK1/2 on GABA_B_ receptors in neurons, we treated primary cortical neuron–glia co-cultures with the potent ERK1/2 inhibitor Ravoxertinib (10 nM) for 10 min. Since the downregulation of GABA_B_ receptors was prevented by inhibition of CaMKII within a time frame of 5–10 min in this pathway [20], we accordingly selected an incubation time of 10 min for the inhibition of ERK1/2. Inhibition of ERK1/2 by Ravoxertinib upregulated total and cell surface expression of GABA_B_ receptors, as shown by immunofluorescence staining for total GABA_B1_ (Figure 2A) and for cell surface GABA_B2_ (Figure 2B). These results were further confirmed by Western blotting, showing that inhibition of ERK1/2 increased the total expression of GABA_B1a_, GABA_B1b_ and GABA_B2_ (Figure 2C).

### 2.2. CaMKIIβ and ERK1/2 Mediate Phosphorylation of GABA_B1_ S867 and T872

Phosphorylation of S867 in GABA_B1_ mediated by CaMKIIβ triggers the sorting of internalized receptors to lysosomal degradation [20,31,32]. A phospho-proteomics study [23] revealed an additional phosphorylation site (T872) adjacent to S867 in GABA_B1_. Since ERK1/2 also downregulated GABA_B_ receptor expression, we hypothesized that ERK1/2 might be involved either in phosphorylating GABA_B1_(S867) or GABA_B1_(T872). Therefore, we first inactivated the phosphorylation sites S867 and T872 in GABA_B1_ by mutating them to alanine (GABA_B1_(S867A) and GABA_B1_(T872A)) and analyzed the cell surface expression of GABA_B_ receptors containing the HA-tagged GABA_B1_ phospho-mutants after transfecting them into cultured neurons. GABA_B_ receptors containing GABA_B1_(S867A) or GABA_B1_(T872A) showed considerably higher cell surface expression than GABA_B_ receptors containing HA-tagged wild-type GABA_B1_ (Figure 3A). Thus, both GABA_B1_ phosphorylation sites are involved in regulating the receptor expression.

Next, we tested whether CaMKIIβ and ERK1/2 are phosphorylating S867 and/or T872 in GABA_B1_. For this, we transfected HEK-293 cells with ERK1, ERK2 or CaMKIIβ together with wild-type GABA_B1_/GABA_B2_ or GABA_B1_(S867A)/GABA_B2_ and evaluated the phosphorylation of GABA_B_ receptors by in situ PLA using antibodies directed against GABA_B1_ and antibodies specifically recognizing phosphorylated serine. As expected, co-expression of CaMKIIβ with GABA_B_ receptors considerably enhanced phosphorylation of the receptors as compared to GABA_B_ receptors expressed in the absence of CaMKIIβ (Figure 3B). As shown previously [20,32], receptors containing the phospho-mutant GABA_B1_(S867A) showed vastly diminished phosphorylation below control levels (wild-type receptors alone) (Figure 3B). Interestingly, co-expression of ERK1 or ERK2 also mediated the phosphorylation of GABA_B_ receptors, which was prevented in receptors containing the phospho-mutant GABA_B1_(S867A) (Figure 3B). Thus, both CaMKIIβ and ERK1/2 appear to mediate the phosphorylation of GABA_B_ receptors at S867 in GABA_B1_.

We then analyzed if CaMKIIβ and ERK1/2 also mediate phosphorylation of GABA_B_ receptors at T872 in GABA_B1_ using the same approach described above except that GABA_B1_(T872A) and antibodies directed against phosphorylated threonine were used. Again, we detected enhanced CaMKIIβ, ERK1 and ERK2-mediated phosphorylation of GABA_B_ receptors, which was blocked in receptors containing GABA_B1_(T872A) (Figure 3C). Hence, CaMKIIβ and ERK1/2 also appear to mediate the phosphorylation of GABA_B_ receptor at T872 in GABA_B1_. These results indicate that both CaMKIIβ and ERK1/2 mediate the phosphorylation of GABA_B_ receptors at S867 and T872 in the GABA_B1_ subunit.

### 2.3. Preventing Phosphorylation of S867 and T872 in GABA_B1_ Upregulates GABA_B_ Receptor Expression

Next, we analyzed the effect of CaMKIIβ and ERK1/2 mediated phosphorylation of S867 and T872 in GABA_B1_ on the expression of GABA_B_ receptors. For this, we used the same experimental setup as described above, but instead of testing phosphorylation, we assessed the cell surface and total expression of the receptors in HEK-293 cells using immunofluorescence staining. Both cell surface (Figure 4A) and total GABA_B_ receptor expression (Figure 4B) were downregulated by co-expressing the receptors with ERK1, ERK2 or CaMKII. However, receptors containing the phosphorylation-deficient mutants GABA_B1_(S867A) or GABA_B1_(T872A) were considerably upregulated as compared to wild-type receptors (Figure 4A,B). This implies that CaMKIIβ and ERK1/2 mediated phosphorylation of S867 and T872 in GABA_B1_ downregulates the GABA_B_ receptor expression.

We then verified the effect of CaMKIIβ and ERK1/2 mediated phosphorylation of GABA_B_ receptors in GABA_B1_ at S867 and T872 on receptor expression in cultured neurons. For this purpose, we transfected cultured neurons with wildtype GABA_B1_/GABA_B2_ or its phosphorylation deficient mutants GABA_B1_(S867A)/GABA_B2_ or GABA_B1_(T872A)/GABA_B2_ and monitored cell surface receptor expression after inhibition of CaMKII with KN93 or blocking ERK1/2 with Ravoxertinib. There was a substantial increase in the cell surface expression of GABA_B_ receptors in neurons by either inhibiting CaMKII or ERK1/2 activity or by preventing phosphorylation of T872 (Figure 4C) or S867 in GABA_B1_ (Figure 4D) by using the phospho-mutants. There was no additive effect of inhibiting CaMKII or ERK1/2 on the expression of receptors containing the phospho-mutants, suggesting that phosphorylation of GABA_B1_(S867) and GABA_B1_(T872) is required and sufficient for inducing degradation of the receptors.

### 2.4. CaMKII Activates ERK1/2 and Is Required for the Interaction of ERK1/2 with GABA_B_ Receptors

Our results described above indicate that both CaMKIIβ and ERK1/2 mediate the phosphorylation of GABA_B1_ at S867 and T872 after co-expression with GABA_B_ receptors in HEK-293 cells. In contrast to CaMKIIβ, ERK1/2 is ubiquitously expressed in non-neuronal cells. It was, therefore, possible that ERK1/2, endogenously expressed in HEK-293 cells, phosphorylates the two sites under the control of CaMKIIβ. This is a very likely scenario as it has been reported that ERK1/2 can be activated by CaMKII, e.g., in epithelial, mesenchymal cells, cardiac myoblasts and neurons [24,25,26,28,29]. To address this issue, we first tested by Western blotting whether HEK-293 cells endogenously express ERK1/2 and CaMKIIβ. As expected, untransfected HEK-293 cells endogenously express ERK1/2 but not CaMKIIβ (Figure 5A). Next, we tested whether ERK1/2 rather than CaMKIIβ is phosphorylating GABA_B_ receptors. For this, HEK-293 cells were transfected with GABA_B_ receptors in the presence or absence of CaMKIIβ and tested for Ser and Thr-specific phosphorylation by in situ PLA after blocking endogenous ERK1/2 activity with Ravoxertinib. Interestingly, basal Ser and Thr phosphorylation of the receptors was significantly inhibited by blocking ERK1/2 activity, indicating that endogenous phosphorylation of GABA_B_ receptors in HEK-293 cells is largely mediated by ERK1/2 (Figure 5B). Co-expression of CaMKIIβ considerably increased phosphorylation of the receptors, which was almost completely inhibited by blocking ERK1/2 activity (Figure 5B). This suggests that ERK1/2 phosphorylation of GABA_B_ receptors is greatly enhanced by CaMKIIβ.

To test if CaMKIIβ enhances the activity of ERK1/2, we treated neuronal cultures with the CaMKII inhibitor KN93 (10 µM for 10 min) and analyzed the ratio of activated ERK1/2 (phosphorylated at Thr183 and Tyr185) over total ERK1/2 expression using phospho-site and pan-specific ERK1/2 antibodies. As expected, blocking CaMKII activity also inhibited the activity of ERK1/2 as tested by immunofluorescence staining (Figure 5C) and by Western blotting (Figure 5D). 

We previously showed that only activated CaMKIIβ interacts with GABA_B_ receptors [20]. Therefore, we analyzed if CaMKII activity is required for the interaction of GABA_B_ receptors with ERK1/2. For this purpose, we treated primary cultured neurons with or without the inhibitors of ERK1/2 (Ravoxertinib) or CaMKII (KN93) and evaluated the interaction of GABA_B_ receptors with ERK1/2 or CaMKIIβ by in situ PLA. Blocking CaMKII activity reduced the interaction of GABA_B_ receptors with CaMKIIβ and ERK1/2 (Figure 5E,F). However, blocking ERK1/2 activity inhibited the interaction of ERK1/2 with GABA_B_ receptors but did not affect the interaction of CaMKIIβ with the receptors, suggesting that activation of ERK1/2 by CaMKIIβ is required for the interaction and thereby phosphorylation of GABA_B_ receptors (Figure 5F). These results indicate that only activated CaMKIIβ and ERK1/2 interact with GABA_B_ receptors. In addition, activation of ERK1/2 by CaMKII appears to be necessary for the interaction of ERK with the receptor. However, the interaction of CaMKIIβ with GABA_B_ receptors appears to be independent of ERK1/2 activity. 

### 2.5. ERK1/2 Activity Is Required for Downregulation of GABA_B_ Receptors under Ischemic Conditions

Under excitotoxic/ischemic conditions, GABA_B_ receptors are downregulated via a mechanism involving phosphorylation of GABA_B1_ at S867 mediated by CaMKIIβ [20,31,32]. To test for a potential contribution of ERK1/2, we subjected cultured neurons to oxygen and glucose deprivation (OGD) stress, treated them afterward with the ERK1/2 inhibitor Ravoxertinib and determined the expression level of GABA_B_ receptors by immunofluorescence staining (Figure 6A,B) and Western blotting (Figure 6C). As observed previously [20,31,32], cell surface and total receptors were downregulated after OGD stress. However, inhibition of ERK1/2 restored total (Figure 6A,C), as well as cell surface expression (Figure 6B) of the receptors. This result implies that phosphorylation of GABA_B_ receptors by ERK1/2 is required for downregulating the receptors under ischemic conditions.

Because CaMKII activates ERK1/2 (Figure 5), we tested for increased interaction of CaMKIIβ with ERK1/2 by in situ PLA in cultured neurons after OGD stress. There were considerably more CaMKIIβ-ERK1/2 interactions after OGD stress as compared to unstressed neurons (Figure 6D). In addition, the CaMKIIβ-ERK1/2 interactions were strongly reduced after blocking CaMKII activity by KN93 in unstressed control neurons and stressed neurons (Figure 6D). This supports the view that CaMKIIβ activates ERK1/2 in OGD-stressed neurons based on their increased interaction.

## 3. Discussion

CaMKIIβ-mediated phosphorylation of GABA_B1_ at S867 is required for sorting the receptors to lysosomal degradation [19,20,32]. The results of this study suggest that CaMKIIβ does not directly phosphorylate GABA_B_ receptors but activates ERK1/2, which in turn phosphorylate GABA_B1_ at S867 and T872 (Figure 7). 

Keeping the number of functional GABA_B_ receptors residing in the plasma membrane at a level that provides adequate neuronal inhibition is fundamental to normal brain function. Under physiological conditions, cell surface expression of GABA_B_ receptors is precisely controlled by constitutive endocytosis, recycling, degradation, and new synthesis of receptors [33]. However, under pathological conditions, GABA_B_ receptors are commonly downregulated, compromising a healthy excitation/inhibition balance [12,13,14,15,16,17,18]. Pathological downregulation of GABA_B_ receptors is associated with inhibition of fast receptor recycling and increased sorting of the receptors to lysosomes for degradation [19,20,21,31,34,35]. The sorting of GABA_B_ receptors to the degradation pathway is controlled by the phosphorylation of S867 in the GABA_B1_ subunit mediated by CaMKIIβ [19,20]. Since a proteomic study analyzing phosphorylated proteins in plasma membranes isolated from mouse cerebellum identified T872 in GABA_B1_ as another phosphorylation site close to GABA_B1_(S867) [23], this raised the question as to whether phosphorylation of GABA_B1_(T872) also is involved in sorting GABA_B_ receptors to lysosomal degradation. In this respect, we found that phosphorylation of both GABA_B1_(S867) and GABA_B1_(T872) is required for receptor degradation. This conclusion is based on the observation that (1) CaMKIIβ mediates phosphorylation of both sites, (2) that mutational inactivation of GABA_B1_(S867) or GABA_B1_(T872) considerably increased GABA_B_ receptor expression, i.e., prevented degradation of the receptors, and (3) that blocking of CaMKII activity did not further increased expression of receptors containing inactivated GABA_B1_(S867) or GABA_B1_(T872) phosphorylation sites. 

Since there is considerable evidence that CaMKII can activate ERK1/2 [24,25,26,27,28,29], we tested whether ERK1/2 is involved in phosphorylating GABA_B1_(S867) and/or GABA_B1_(T872). As we previously observed for CaMKIIβ [20,32], overexpression of ERK1 or ERK2 considerably reduced the expression of GABA_B_ receptors, whereas blocking ERK activity increased their expression. Like CaMKIIβ, ERK1 and ERK2 also mediate phosphorylation of GABA_B1_(S867) and GABA_B1_(T872), and mutational inactivation of either site significantly increased GABA_B_ receptor expression, which was not further increased by blocking ERK activity. Inhibition of ERK activity also restored GABA_B_ receptor expression after subjecting neurons to ischemic stress. In addition, ischemic stress increased the interaction of CaMKIIβ with ERK1/2, which was inhibited by blocking CaMKII activity. Therefore, both CaMKIIβ and ERK1/2 appear to be part of the same pathway involved in GABA_B_ receptor downregulation and degradation.

Our observation that blocking CaMKII activity also reduced ERK activity suggests that CaMKIIβ does not directly phosphorylate GABA_B_ receptors but activates ERK1/2, which then phosphorylate GABA_B1_(S867) and GABA_B1_(T872). This conclusion is supported by our finding that (1) blocking the activity of endogenously expressed ERK1/2 in HEK-293 cells almost completely prevented basal Ser and Thr phosphorylation of the receptors and, most strikingly, CaMKIIβ-induced phosphorylation of the receptors; (2) that the interaction of GABA_B_ receptors with ERK1/2 is dependent on the activity of CaMKII, whereas the interaction of CaMKII with the receptors is independent of ERK1/2 activity; (3) that downregulated GABA_B_ receptor expression after ischemic stress in cultured neurons is completely restored by blocking solely ERK1/2 activity; and (4) that the enhanced interaction of CaMKIIβ with ERK1/2 after ischemic stress is inhibited by blocking CaMKII activity.

Concerning the mechanism of ERK1/2 activation by CaMKIIβ, it is unlikely that CaMKIIβ directly phosphorylates ERK1/2. The canonical activation of ERK1/2 is a three-stage process involving the proteins Ras, Raf and MEK. Extracellular stimuli activate the GTPase Ras by binding GTP. Ras-GTP then binds to and activates the kinase RAF, which in turn phosphorylates and activates MEK (MAPK/ERK1/2 kinase). Activated MEK finally phosphorylates and thereby activates ERK1/2. However, Raf can also be activated by intracellular kinases, such as protein kinase A, protein kinase C or CaMKII. Previous reports have shown that CaMKII mediates activation of ERK1/2 not directly but by phosphorylating and thereby activating Raf, which then initiates the ERK1/2 activation cascade [24,26,28]. In this sense, CaMKIIβ most likely recruits and activates the ERK1/2 signaling complex to GABA_B_ receptors and thereby induce phosphorylation of GABA_B1_(S867) and GABA_B1_(T872) rather than contributing directly to their phosphorylation. The recruitment of ERK1/2 to GABA_B_ receptors by CaMKIIβ is supported by our in situ PLA experiments, indicating that blocking of CaMKII activity inhibits the interaction of ERK1/2 with the receptors.

We had previously shown that blocking the interaction of CaMKIIβ with GABA_B_ receptors after an ischemic insult restored downregulated GABA_B_ receptor expression and function to normal levels. This diminished ischemia-induced neuronal overexcitation and inhibited progressive neuronal death [32]. Like CaMKIIβ, ERK1/2 is crucial for the pathological downregulation of GABA_B_ receptors under ischemic conditions and a potential candidate for the development of a novel neuroprotective therapy. Since ERK1/2, like CaMKII, is involved in numerous cellular functions and can promote cell death, as well as survival pathways [30,36,37,38], globally inhibiting ERK activity after an ischemic insult might not be an ideal approach for a neuroprotective strategy. Instead, inhibiting the interaction of ERK1/2 with GABA_B_ receptors using interfering peptides might be a promising, highly specific approach for the development of a neuroprotective treatment in cerebral ischemia or other neurological diseases associated with neuronal overexcitation and downregulation of GABA_B_ receptors.

## 4. Materials and Methods

### 4.1. Drugs

The following drugs were used for this study: Ravoxertinib (10 nM, Selleck Chemicals LLC, via LuBioScience, Zurich, Switzerland.) and KN93 (10 μM, Sigma-Aldrich, Buchs, Switzerland). 

### 4.2. Plasmids

The following plasmids were used for this study: HA-tagged GABA_B1a_ [39]; GABA_B2_ [3]; HA-tagged GABA_B1a_(S867A) and HA-tagged GABA_B1a_(T872A) (mutations were custom-made by GenScript, Piscataway NJ, USA), ERK1 (Addgene plasmid 12656, Watertown, MA, USA) [40], ERK2 (Addgene plasmid 40812) [40] and GFP-tagged CaMKIIβ (Addgene plasmid 21227) [41].

### 4.3. Antibodies

Mouse CaMKIIβ (1:1500 for Western blotting [WB] and 1:150 for in situ proximity ligation assay [PLA], ThermoFisher Scientific #13-9800, Basel, Switzerland), mouse GABA_B1_ (1:250 for immunofluorescence staining [IF] and, 1:100 for PLA; Abcam #ab55051), rabbit GABA_B1bN_ directed against the N-terminus of GABA_B1b_ (affinity-purified,1:100 for IF; custom made by GenScript), rabbit GABA_B2N_ directed against the N-terminus of GABA_B2_ (affinity-purified, used for cell surface staining, 1:25 for IF; custom made by GenScript), rabbit GABA_B2_ (1:500 for IF, 1:100 for PLA, 1:800 for Western blotting; Abcam #ab75838, via Lucerna-Chem AG, Luzern, Switzerland), mouse phospho-serine (1:150 for PLA, Sigma-Aldrich #P5747), mouse phospho-threonine (1:150 for PLA, Millipore #708014, Darmstadt, Germany), rabbit ERK1/2 (1:500 for WB and IF, 1:100 for PLA, Sigma-Aldrich #M5670), mouse pERK1/2 (1:250 for WB and IF, Sigma-Aldrich #M9692), rabbit HA (1:500 for IF, Sigma-Aldrich #SAB5600116). For immunofluorescence staining, secondary antibodies used were labeled with Alexa Fluor Plus 488, 555, or 647 (1:2000, ThermoFisher), and for Western blotting, antibodies were conjugated to IRDye 700CW or IRDye800CW (LI-COR Biosciences, Bad Homburg, Germany).

### 4.4. Neuron–Glia Co-Cultures

The use of rat embryos and the procedure for generating primary neuronal cultures was approved by the Zurich cantonal veterinary office, Zurich, Switzerland (license ZH011/19 and ZH087/2022). All cell culture media used were from Gibco. Neuron–glia co-cultures were prepared according to Buerli et al. [42]. The cerebral cortexes of 18-day-old rat embryos were carefully dissected and washed with 5 mL sterile-filtered PBGA buffer (PBS containing ten mM glucose, 1 mg/mL bovine serum albumin and antibiotic-antimycotic 1:100 [10,000 units/mL penicillin; 10,000 μg/mL streptomycin; 25 μg/mL amphotericin B]). The cortices were cut into small pieces with a sterile scalpel and digested in 5 mL sterile filtered papain solution for 15 min at 37 °C. The supernatant was removed, and the tissue was washed twice with complete DMEM/FCS medium (Dulbecco’s Modified Eagle’s Medium containing 10% Fetal Calf Serum and penicillin/streptomycin, 1:100). Then, fresh DMEM/FCS was added, and the tissue was gently triturated and subsequently filtered through a 40 μm cell-strainer. Finally, the cells were plated at a concentration of 70,000–90,000 per well onto the poly L-lysine (50 μg/mL in PBS) coated coverslips in a 12-well culture dish and incubated overnight at 37 °C and 5% CO_2_. After 24 h of incubation, the DMEM medium was exchanged with freshly prepared NU-medium (Minimum Essential Medium (MEM) with 15% NU serum, 2% B27 supplement, 15 mM HEPES, 0.45% glucose, 1 mM sodium pyruvate, 2 mM GlutaMAX). The cultures were kept for 12–16 days in vitro.

### 4.5. Transfection of Neurons

Primary cultures (co-cultures of neurons and glia cells) were used for transfection with plasmids after 11–12 days in culture. Plasmid DNA was transfected into neurons by magnetofection using Lipofectamine 2000 (Invitrogen, Waltham, WA, USA) and CombiMag (OZ Biosciences, San Diego, CA, USA) as specified by Buerli et al. [42].

### 4.6. Culture and Transfection of HEK-293 Cells

HEK-293 cells (Human Embryonic Kidney, ATCC) were cultured in DMEM (Gibco Life Technologies, via ThermoFisher Scientific, Basel, Switzerland) containing 10% fetal bovine serum (Gibco Life Technologies) and penicillin/streptomycin (Gibco Life Technologies). HEK-293 cells were transfected with plasmids using the polyethyleneimine (PEI) method according to the jet-PEI protocol (Polyplus Transfection, via Brunschwig, Basel, Switzerland).

### 4.7. Oxygen and Glucose Deprivation (OGD) Stress

The OGD medium (DMEM lacking glucose, glutamine, sodium pyruvate, HEPES and phenol red) was deprived of oxygen by equilibrating it with nitrogen for 15 min in a water bath at 37 °C. For subsequent immunofluorescence staining, 1 mL of equilibrated OGD medium was added to each well of a 12-well culture plate, and the coverslips containing the cultured neurons were transferred to the OGD medium. The culture plate was then incubated for 1 h in a hypoxic incubator at 1% O_2_, 5% CO_2_ and 37 °C. The coverslips were then transferred back to the culture plate containing the original conditioned culture medium and incubated at 37 °C and 5% CO_2_.

For Western blotting experiments, neurons/glia were cultured in 6 cm dishes. OGD stress was induced by removing (and saving) the culture medium and adding 5 mL of equilibrated OGD medium to the cultures, followed by 1 h incubation in a hypoxic incubator at 1% O_2_, 5% CO_2_ and 37 °C. The OGD medium was then removed, 5 mL of the saved original conditioned culture medium was added, and the cultures were incubated at 37 °C and 5% CO_2_. The neurons were analyzed after a recovery period of 10 min.

### 4.8. Immunofluorescence Staining

For cell surface staining of GABA_B_ receptors, an antibody directed against the extracellularly located N-terminus of GABA_B1b_ or GABA_B2_ (GABA_B1bN_ and GABA_B2N_) was used. Coverslips containing the cultured neuron/glia cells were washed 3 times with cold buffer A (25 mM HEPES pH 7.4, 119 mM NaCl, 2.5 mM KCl, 2 mM CaCl_2_, 1 mM MgCl_2_ and 30 mM glucose). Then, the GABA_B2N_ antibody (1:250 dilution in buffer A containing 10% normal donkey serum (NDS)) was added and incubated on ice for 90 min. The coverslips were then washed 3 times for 5 min with buffer A, followed by incubation with Alexa Fluor Plus anti-rabbit secondary antibody (1:2000 in PBS/10% NDS, ThermoFisher, #A32790) for 60 min on ice. Afterwards, the coverslips were washed again 3 times for 5 min with buffer A. For subsequent staining of total GABA_B1_, the cells were fixed with 4% PFA for 30 min at room temperature. After fixation, the cells were washed with PBS and permeabilized by incubation for 12 min in 0.2% Triton X-100/PBS. Then, GABA_B1_ antibody (1:250, Cat No. ab55051, Abcam, via Lucerna-Chem AG, Luzern, Switzerland) was added and incubated overnight at 4 °C. After incubation, the coverslips were washed 3 times for 5 min with PBS and incubated with Alexa Fluor Plus anti-mouse secondary antibody (dilution 1:2000 in PBS/10% NDS, ThermoFisher, #A32773) for 1 h at room temperature. Finally, the coverslips were washed again 3 times for 5 min with PBS and mounted in DAKO fluorescence mounting medium onto glass slides for confocal microscopy.

### 4.9. In Situ Proximity Ligation Assay (In Situ PLA)

The in situ PLA was used to investigate the phosphorylation of GABA_B1_ by CaMKII and ERK1/2 using antibodies against GABA_B1_ (1:50, Cat No. ab55051, Abcam) or ERK-1/2 (1:100, Sigma-Aldrich #M5670) and phospho-serine (1:150, Sigma-Aldrich #P5747) or phospho-threonine (1:150 for PLA, Millipore #708014). In addition, the interaction of GABA_B1_ with CaMKIIβ (antibody 1:150, ThermoFisher Scientific #13-9800) and ERK1/2 (antibody 1:100, Sigma-Aldrich #M5670) was analyzed. The in situ PLA was performed using the Duolink II kit (Sigma Aldrich) according to the instructions of the manufacturer. Briefly, the neurons were washed for 5 min with PBS and then fixed with 4% PFA for 30 min at room temperature. Then, the coverslips were rinsed in PBS for 5 min and permeabilized for 15 min with 0.2% Triton X-100/PBS. After rinsing the coverslips in PBS, they were incubated with the two primary antibodies of interest (one antibody raised in rabbit and other in mouse) and kept in a humidity chamber overnight at 4 °C. Subsequently, the cultures were washed four times for 5 min with PBS and incubated for 20–30 min at room temperature with the PLA probes (prepared by diluting anti-Mouse MINUS and anti-Rabbit PLUS (Duolink II) 1:5 in 5% BSA/PBS). Afterward, 40 μL of the PLA probe solution was pipetted on top of each coverslip and incubated in a humidity chamber for 1 h at 37 °C. The coverslips were then washed two times for 5 min in PBS and incubated for 1 h at 37 °C with ligation solution. Subsequently, the cells were washed two times in Duolink II Wash Buffer A and incubated with the amplification solution supplemented with Alexa Fluor 488 Plus anti-mouse secondary antibody (for detection of GABA_B1_) at 37 °C for 100 min. Finally, the coverslips were washed two times for 10 min with Duolink II Wash Buffer B in the dark and mounted onto microscope slides with DAKO fluorescent mounting medium.

### 4.10. Western Blotting

For Western blot analysis, neuron–glia co-cultures were grown for 12 days on 6 cm culture dishes plated with 500,000 cells. Cultures were washed two times with ice-cold PBS, harvested, and homogenized by sonication, followed by a determination of protein content using the Bradford protein assay (BioRad, Hercules, CA, USA). The samples were incubated with Laemmli sample buffer (Bio-Rad) for 1 h at 37 °C and aliquots containing 25 μg protein were subjected to sodium dodecyl sulfate-polyacrylamide gel electrophoresis (SDS-PAGE) using 7.5% (for GABA_B_ receptor subunits) or 10% (for ERK1/2) mini-gels (Mini Protean3; Bio-Rad). Proteins were transferred onto nitrocellulose membranes in a semi-dry transfer cell (Trans-Blot SD; Bio-Rad) at 15 V for 75 min. After blotting, the transferred total proteins were stained with REVERT 700 Total Protein Stain (LI-COR Biosciences) and detected by the ODYSSEY CLx scanner (LI-COR Biosciences). After de-staining, the blots were blocked for 1 h in PBS containing 5% nonfat dry milk at room temperature, followed by incubation with primary antibody overnight at 4 °C in PBS containing 5% nonfat dry milk. The blots were then washed five times for 5 min with TBST and incubated with secondary antibodies for 1 h at room temperature. The blots were washed again with TBST, and immunoreactivity was detected by the ODYSSEY CLx scanner (LI-COR Biosciences). Immunoreactivity was quantified with the Image Studio software (version 5.2.5, LI-COR Biosciences) and normalized to total protein in the corresponding lanes.

### 4.11. Microscopy and Image Analysis

Images were taken with a Zeiss laser scanning confocal microscope (CLSM800) equipped with a Zeiss 63× (1.4 NA) plan-apochromatic oil differential interference contrast objective in sequential mode with a resolution of 1024 × 1024 pixels. The laser intensity and the detector gain were adjusted to values that avoid signal saturation, and all images of one experiment were imaged with the same settings in one continuous session. The images were quantitatively analyzed using the ImageJ software (version 2.3.0).

For quantification of cell surface staining, the outer and inner perimeter of the cell surface were exactly outlined. Then, the fluorescence intensity value obtained from the inner border was subtracted from the one of the outer border so that only the fluorescence present at the cell surface was determined. For quantification of the total cell staining, the outer border of the cell was marked, and the mean fluorescence intensity was measured.

For quantification of in situ PLA signals, the soma of neurons or HEK-293 cells was surrounded, and the fluorescent dots inside these borders were counted using the ImageJ option “Find maxima”. A fixed noise tolerance value was used for the analysis of all images of the same experiment. The PLA signals were normalized to the area analyzed and to the GABA_B_ receptor expression level.

### 4.12. Statistics

The statistical evaluation of data was performed using the software GraphPad Prism (version 8.4.3). All results were given as mean value ± standard deviation (SD). The data were analyzed by unpaired two-tailed *t*-test, one-way or two-way ANOVA followed by appropriate post hoc tests as indicated in the figure legends. Data sets were tested for normal or lognormal distributions. In case of significant deviation from homoscedasticity, Welch and Brown Forsythe variations of ANOVA were used. A *p*-value of <0.05 was considered statistically significant.

## Figures and Tables

**Figure 1 ijms-24-13436-f001:**
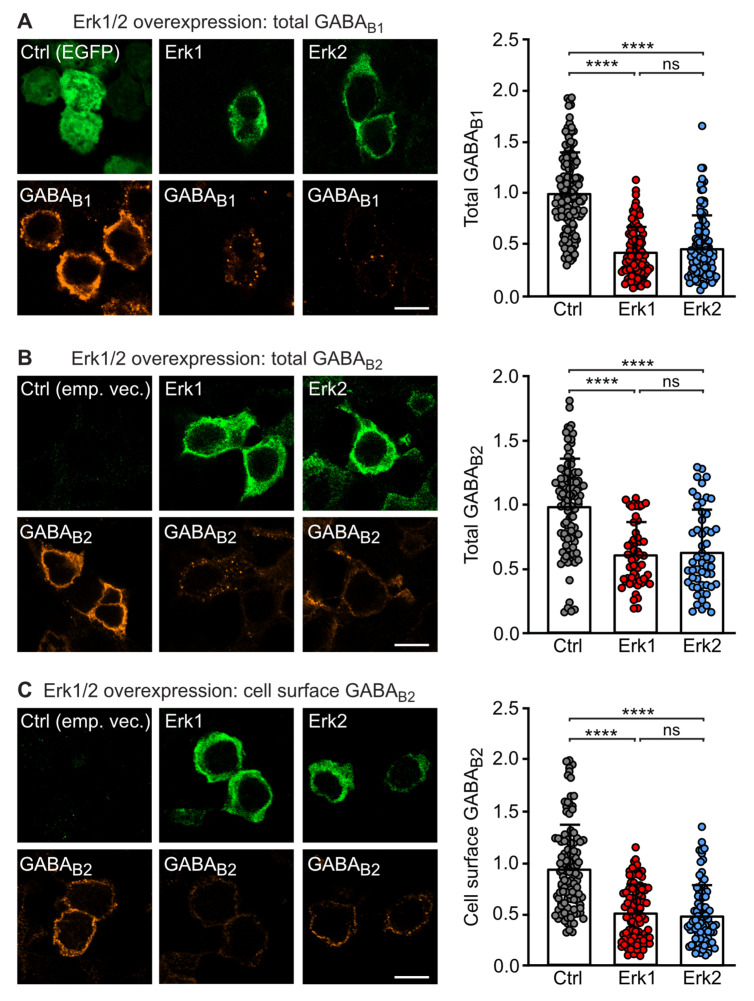
Overexpression of ERK1/2 downregulates GABA_B_ receptors. HEK 293 cells were transfected with GABA_B1_, GABA_B2_ and empty vector or EGFP as a control (Ctrl) or with GABA_B1_, GABA_B2_ and either ERK1 or ERK2. After 2 days, the cells were tested for ERK1/2 and GABA_B_ receptor expression by immunofluorescence staining using antibodies directed against ERK1/2 and GABA_B1_ or GABA_B2_. (**A**) Transfection with ERK1 or ERK2 downregulated the expression of total GABA_B1_. **Left**: representative images (scale bar: 10 µm). **Right**: quantification of fluorescence intensities. The data represent the mean ± SD of 101–148 cells per condition from four independent experiments. Brown–Forsythe and Welch’s ANOVA with Games-Howell’s post-test (ns, *p* > 0.05; ****, *p* < 0.0001). (**B**) Transfection with ERK1 or ERK2 downregulated the expression of total GABA_B2_. **Left**: representative images (scale bar: 10 µm). **Right**: quantification of fluorescence intensities. The data represent the mean ± SD of 54–95 cells per condition from four independent experiments. One Way ANOVA with Tukey’s post-test (ns, *p* > 0.05; ****, *p* < 0.0001). (**C**) Transfection with ERK1 or ERK2 downregulated the cell surface expression of GABA_B_ receptors as probed with antibodies directed against an extracellular located epitope in the N-terminal domain of GABA_B2_. **Left**: representative images (scale bar: 10 µm). **Right**: quantification of fluorescence intensities. The data represent the mean ± SD of 84–113 cells per condition from four independent experiments. Brown–Forsythe and Welch’s ANOVA with Games-Howell’s post-test (ns, *p* > 0.05; ****, *p* < 0.0001).

**Figure 2 ijms-24-13436-f002:**
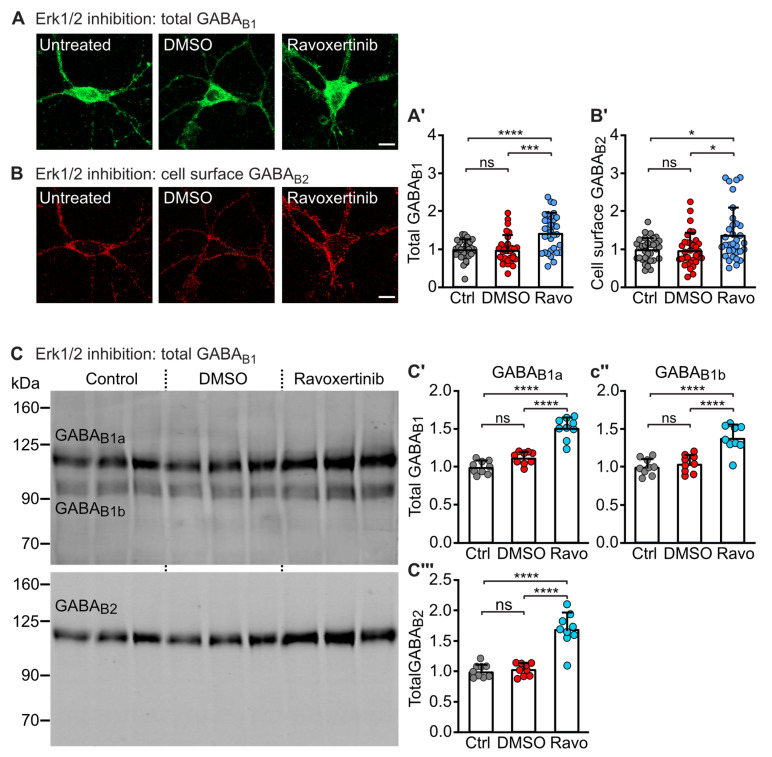
ERK1/2 regulates GABA_B_ receptor expression in cultured neurons. Neurons were treated or not for 10 min with the ERK1/2 inhibitor Ravoxertinib (10 nM) or with DMSO (1% final concentration, as a control for the effect of the solvent used for dissolving Ravoxertinib) and then tested for GABA_B_ receptor expression. (**A**) Treatment with Ravoxertinib upregulated the expression of total GABA_B_ receptors as determined using antibodies directed against GABA_B1_. **Left** (**A**): representative images (scale bar: 10 μm). **Right** (**A’**): quantification of fluorescence intensities. Signals were normalized to control (cultures not treated with Ravoxertinib). The data represent the mean ± SD of 33 neurons per condition from three independent experiments. Brown–Forsythe and Welch’s ANOVA with Dunnett’s T3 post test (ns, *p* > 0.05; ***, *p* < 0.0005; ****, *p* < 0.0001). (**B**) Treatment with Ravoxertinib upregulated the cell surface expression of GABA_B_ receptors as determined using antibodies directed against an extracellular located epitope in the N-terminal domain of GABA_B2_. **Left** (**B**): representative images (scale bar: 10 μm). **Right** (**B’**): quantification of fluorescence intensities. Signals were normalized to control (cultures not treated with Ravoxertinib). The data represent the mean ± SD of 33 neurons per condition from three independent experiments. Brown–Forsythe and Welch’s ANOVA with Dunnett’s T3 post-test (ns, *p* > 0.05; *, *p* < 0.05). (**C**) Treatment with Ravoxertinib upregulated total GABA_B_ receptor expression as tested by Western blotting using antibodies directed against GABA_B1_ and GABA_B2_. (**C’**–**C’’’**): quantification of Western blot signals. Signals were normalized to control (cultures not treated with Ravoxertinib). Cultures treated with DMSO (1% final concentration) served as controls for the effect of the solvent used for dissolving Ravoxertinib. The data represent the mean ± SD of 9 cultures per condition from three independent neuron preparations. One-way ANOVA with Tukey’s post-test (ns, *p* > 0.05; ****, *p* < 0.0001).

**Figure 3 ijms-24-13436-f003:**
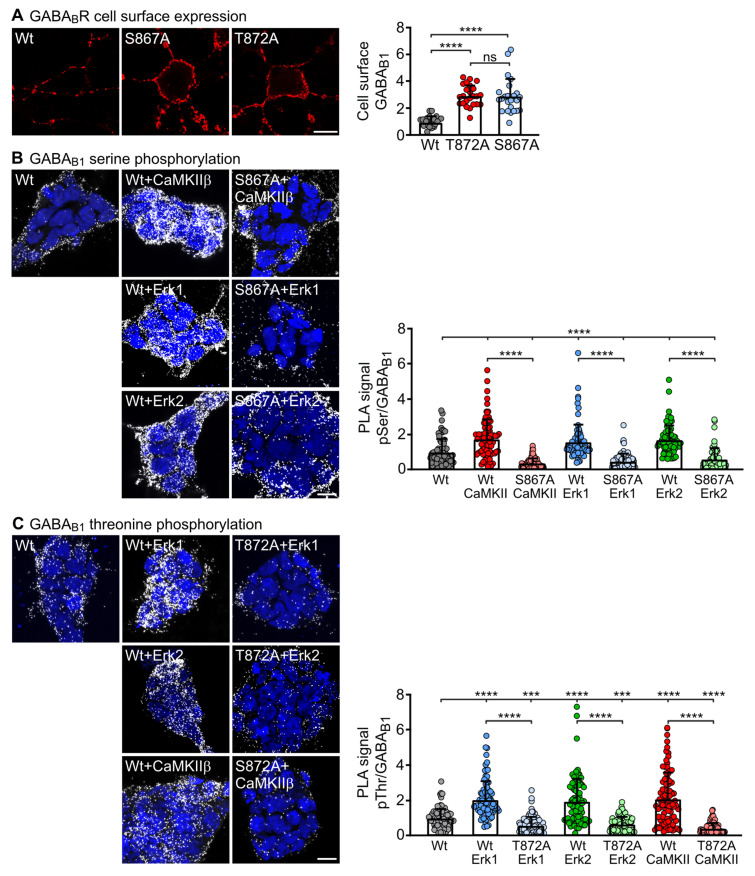
ERK1/2 and CaMKIIβ mediated phosphorylation of GABA_B1_ at serine 867 (S867) and threonine 872 (T872). (**A**) Transfection of neurons with mutant GABA_B1_ containing inactivated phosphorylation sites T872A or S867A upregulated the cell surface expression of GABA_B_ receptors as determined by immunofluorescence staining using antibodies directed against the HA-tagged GABA_B1_ phospho-mutants. **Left**: representative images (scale bar: 10 µm). **Right**: quantification of fluorescent intensities. The data represent the mean ± SD of 24 cells per condition from three independent experiments. Brown–Forsythe and Welch’s ANOVA with Games-Howell’s post-test (ns, *p* > 0.05; ****, *p* < 0.0001). (**B**) ERK1/2 and CaMKII mediated phosphorylation of S867 in GABA_B1_. HEK-293 cells were transfected with wild-type GABA_B1_/GABA_B2_ (wt) or with the phospho-mutant GABA_B1_(S867A)/GABA_B2_ with or without CaMKIIβ, ERK1 or ERK2 and tested for GABA_B_ receptor phosphorylation by in situ PLA using antibodies directed against phospho-serine and HA-tagged GABA_B1_. Signals were normalized to the in situ PLA signals in HEK-cells transfected with wild-type GABA_B1_/GABA_B2_, which served as control. **Left**: representative images; in situ PLA signals (white dots) represent serine phosphorylated GABA_B_ receptors (scale bar: 10 µm). **Right**: quantification of in situ PLA signals. The data represent the mean ± SD of 78 cells per condition from three independent preparations. Brown–Forsythe and Welch’s ANOVA with Dunnett’s T3 multiple comparison test (****, *p* < 0.0001). The upper line depicts the statistical evaluation between the wild-type control and all other conditions. (**C**) ERK1/2 and CaMKII mediated phosphorylation of T872 in GABA_B1_. HEK-293 cells were transfected with wild-type GABA_B1_/GABA_B2_ (wt) or with the phospho-mutant GABA_B1_(T872A)/GABA_B2_ with or without CaMKIIβ, ERK1 or ERK2 and tested for GABA_B_ receptor phosphorylation by in situ PLA using antibodies directed against phospho-threonine and HA-tagged GABA_B1_. Signals were normalized to the in situ PLA signals in HEK cells transfected with wild-type GABA_B1_/GABA_B2_, which served as control. **Left**: representative images; in situ PLA signals (white dots) represent threonine phosphorylated GABA_B_ receptors (scale bar: 10 µm). **Right**: quantification of in situ PLA signals. The data represent the mean ± SD of 86 cells per condition from three independent preparations. Brown–Forsythe and Welch’s ANOVA with Dunnett’s T3 multiple comparison test (***, *p* < 0.0005; ****, *p* < 0.0001). The upper line depicts the statistical evaluation between the wild-type control and all other conditions. PLA, proximity ligation assay.

**Figure 4 ijms-24-13436-f004:**
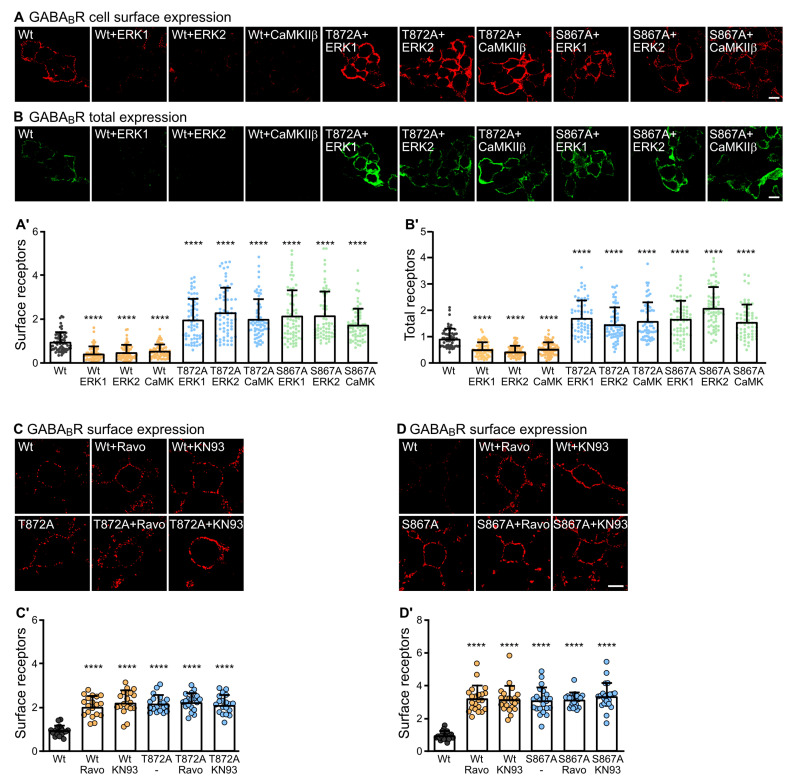
Effect of ERK1/2 and CaMKIIβ mediated phosphorylation of GABA_B1_ T872 and S867 on GABA_B_ receptor expression. (**A**,**B**) HEK 293 cells were transfected with wild-type GABA_B1_/GABA_B2_ (wt) or with the phospho-mutant GABA_B1_(T872A) plus GABA_B2_ or GABA_B1_(S867A) plus GABA_B2_ with or without CaMKIIβ, ERK1 or ERK2 and tested for total and cell surface expression of GABA_B_ receptors by fluorescence staining using antibodies directed against GABA_B1_ for total receptor staining and antibodies directed against an extracellular located epitope in the N-terminal domain of GABA_B2_ for cell surface receptor staining. Co-transfection with GABA_B1_(S867A) or GABA_B1_(T872A) upregulated the cell surface (**A**) and total (**B**) expression of GABA_B_ receptors. **Top**: representative images (scale bar: 10 µm). **Bottom** (**A’**,**B’**): quantification of fluorescence intensities. The data represent the mean ± SD of 62–68 cells per condition from three independent experiments. Brown–Forsythe and Welch’s ANOVA with Games-Howell’s post-test (****, *p* < 0.0001). (**C**) The cell surface expression of GABA_B_ receptors containing the phospho-mutant GABA_B1_(T872A) was unaffected by CaMKII and ERK1/2 inhibitors. Neurons were transfected with wild-type HA-tagged GABA_B1_ or the phospho-mutant HA-tagged GABA_B1_(T872A), treated or not for 10 min with the CaMKII inhibitor KN93 or ERK1/2 inhibitor Ravoxertinib and tested for cell surface expression of GABA_B_ receptors by immunofluorescence staining using antibodies directed against the extracellularly located HA-tag. **Top** (**C**): representative images (scale bar: 10 µm). **Bottom** (**C’**): quantification of fluorescence intensities. The data represent the mean ± SD of 19 cells per condition from three independent experiments. Brown–Forsythe and Welch’s ANOVA with Games-Howell’s post-test (****, *p* < 0.0001). (**D**) The cell surface expression of GABA_B_ receptors containing the phospho-mutant GABA_B1_(T867A) was unaffected by CaMKII and ERK1/2 inhibitors. Neurons were transfected with wild-type HA-tagged GABA_B1_ or the phospho-mutant HA-tagged GABA_B1_(T867A), treated or not for 10 min with the CaMKII inhibitor KN93 or ERK1/2 inhibitor Ravoxertinib and tested for cell surface expression of GABA_B_ receptors by immunofluorescence staining using antibodies directed against the extracellularly located HA-tag. **Top** (**D**): representative images (scale bar: 10 µm). **Bottom** (**D’**): quantification of fluorescence intensities. The data represent the mean ± SD of 22 cells per condition from three independent experiments. Brown–Forsythe and Welch’s ANOVA with Games-Howell’s post-test (****, *p* < 0.0001).

**Figure 5 ijms-24-13436-f005:**
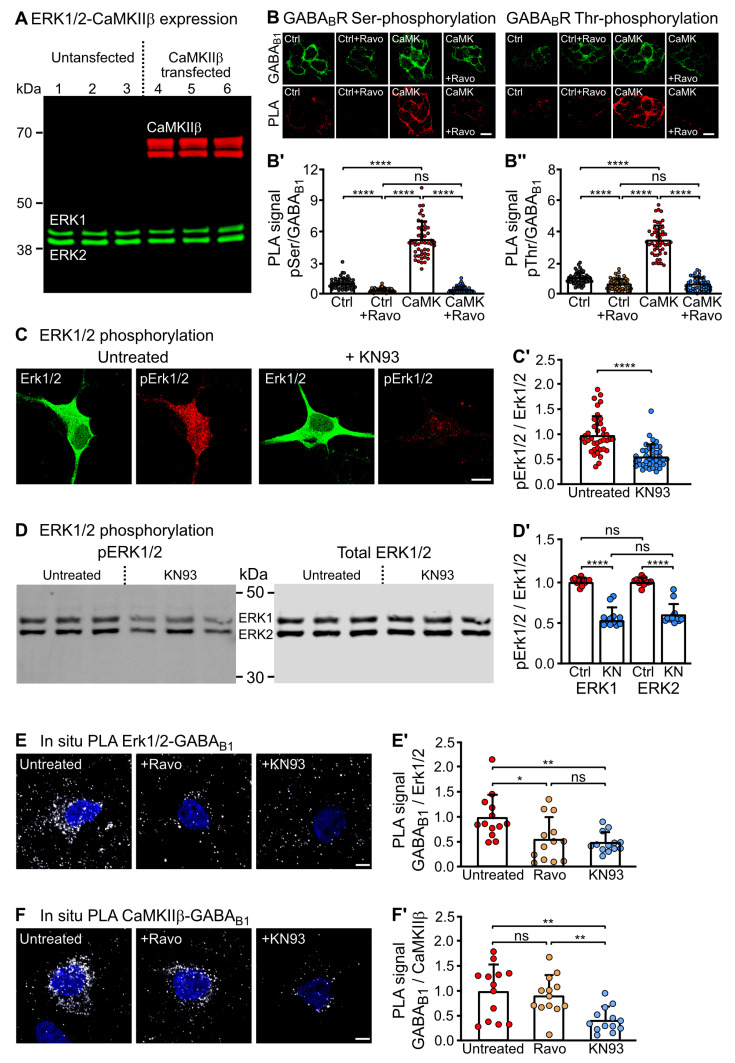
ERK1/2 was activated by CaMKII and interacted activity-dependently with GABA_B_ receptors. (**A**) HEK-293 cells endogenously express ERK1/2 but not CaMKIIβ. Homogenates of untransfected HEK-293 cells were probed for the presence of ERK1/2 and CaMKIIβ by Western blot analysis (lanes 1–3 depict three homogenate preparations). As a control, HEK-293 cells were transfected with CaMKIIβ plasmid (lanes 4–6). (**B**) ERK1/2 but not CaMKIIβ directly phosphorylates GABA_B_ receptors. HEK-293 cells were transfected with GABA_B1_ and GABA_B2_ (Ctrl, Ctrl + Ravo) or with GABA_B1_ and GABA_B2_ plus CaMKIIβ (CaM, CaMK + Ravo) and tested for Ser and Thr phosphorylation by in situ PLA after blocking ERK1/2 activity with 10 nM Ravoxertinib for 10 min (+Ravo) or not. **Top** (**B**): representative images (in situ PLA signals: red dots, scale bar: 10 µm). **Bottom** (**B’**,**B’’**): quantification of in situ PLA signals. The data represent the mean ± SD of 50 cells per condition from two independent experiments. Brown–Forsythe and Welch’s ANOVA with Dunnets’s T3 post-test (ns, *p* > 0.05; ****, *p* < 0.0001). (**C**,**D**) CaMKII activated ERK1/2. Neurons were treated for 10 min with the CaMKII inhibitor KN93 and then tested for activated ERK1/2 (pERK1/2, phosphorylated at Thr183 and Tyr185) and total ERK1/2 expression by immunofluorescence staining (**C**) or Western blotting (**D**) using phospho-specific and pan ERK1/2 antibodies. Inhibition of CaMKII by KN93 also inhibited the activity of ERK1/2. **Left** (**C**,**D**): representative images (scale bar: 10 µm). **Right** (**C’**,**D’**): quantification of fluorescence intensities. (**C**) The data represent the mean ± SD of 39 neurons per condition from three independent experiments. Unpaired two-tailed *t*-test (****, *p* < 0.0001). (**D**) The data represent the mean ± SD of 12 cultures per condition from four independent neuron preparations. One-way ANOVA with Tukey’s post-test (ns, *p* > 0.05; ****, *p* < 0.0001). (**E**) The interaction between GABA_B_ receptors and ERK1/2 in neurons was reduced by inhibition of CaMKII (KN93) or ERK1/2 (Ravoxertinib) as determined by in situ PLA using antibodies directed against GABA_B1_ and ERK1/2. PLA signals were normalized to the untreated control. **Left** (**C**): representative images of in situ PLA signals (white dots, scale bar: 5 µm). **Right** (**E’**): quantification of in situ PLA signals. The data represent the mean ± SD of 13 neurons per condition from two independent experiments. Brown–Forsythe and Welch’s ANOVA with Games-Howell’s post-test (ns, *p* > 0.05; *, *p* < 0.05; **, *p* < 0.01). (**F**) The interaction of CaMKIIβ with GABA_B_ receptors is independent of ERK1/2 activity. Neurons were either untreated or treated with the CaMKII inhibitor KN93 or the ERK1/2 inhibitor Ravoxertinib for 10 min and subsequently analyzed for the interaction of CaMKIIβ with GABA_B_ receptors using antibodies directed against CaMKIIβ and GABA_B1_. Signals were normalized to the untreated control. **Left** (**F**): representative images of in situ PLA signals (white dots, scale bar: 5 µm). **Right** (**F’**): quantification of in situ PLA signals. The data represent the mean ± SD of 13 neurons per condition from two independent experiments. Brown–Forsythe and Welch’s ANOVA with Games-Howell’s post-test (ns, *p* > 0.05; **, *p* < 0.01).

**Figure 6 ijms-24-13436-f006:**
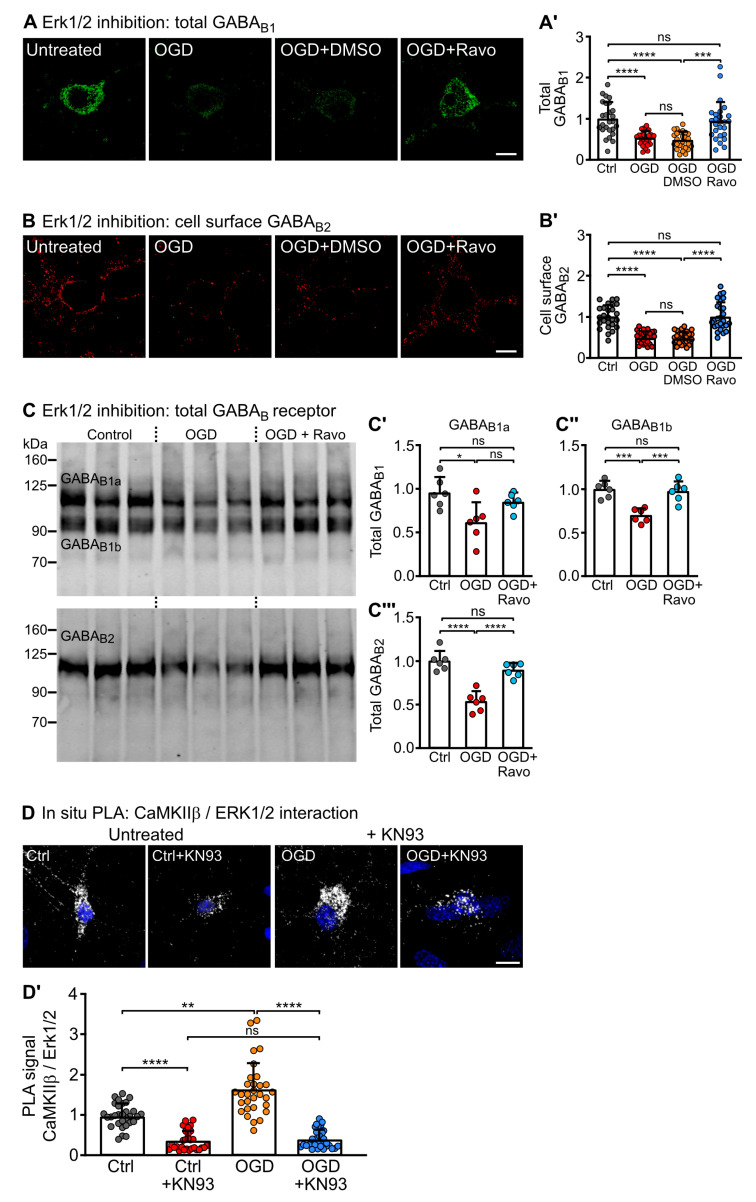
ERK1/2 is involved in ischemic stress-induced downregulation of GABA_B_ receptors. (**A**,**B**) Inhibition of ERK1/2 normalized GABA_B_ receptor total (**A**) and cell surface (**B**) expression after oxygen and glucose deprivation (OGD) induced stress. Neurons were subjected to 1 h of OGD, then treated for 10 min with the ERK1/2 inhibitor Ravoxertinib (10 nM) and analyzed for GABA_B_ receptor expression using antibodies directed against GABA_B1_ (**A**) and GABA_B2_ (**B**) by immunofluorescence staining. OGD downregulated GABA_B_ receptors, and treatment with Ravoxertinib restored normal expression levels. **Right** (**A**,**B**): representative images (scale bar: 10 µm). **Left** (**A’**,**B’**): quantification of fluorescence intensities. The data represent the mean ± SD of 26 neurons per condition from three independent experiments. Brown–Forsythe and Welch’s ANOVA with Dunnett’s T3 post test (ns, *p* > 0.05; ***, *p* < 0.0005; ****, *p* < 0.0001). (**C**) OGD-induced downregulation of GABA_B_ receptor expression was recovered by inhibition of ERK1/2 with Ravoxertinib as verified by Western Blotting. (**C’–C’’’**): quantification of Western blot signals. Signals were normalized to untreated control cultures (control). The data represent the mean ± SD of 3 independent neuron preparations per condition and one technical replicate. One-way ANOVA with Tukey’s post-test (ns, *p* > 0.05; ***, * *p* < 0.05; *p* < 0.0005; ****, *p* < 0.0001). (**D**) The interaction between CaMKIIβ and ERK1/2 in neurons was increased after OGD and blocked by inhibition of CaMKII activity. Neurons were subjected to 1 h of OGD, then treated for 10 min with the CaMKII inhibitor KN93 (10 µM) and analyzed for the interaction of CaMKIIβ and ERK1/2 by in situ PLA. Signals were normalized to untreated cultures (control). **Top** (**D**): representative images of in situ PLA signals (white dots, scale bar: 10 µm). **Bottom** (**D’**): quantification of in situ PLA signals. The data represent the mean ± SD of 30 neurons per condition from three independent experiments. Two-way ANOVA with Tukey’s multiple comparison test (ns, *p* > 0.05; **, *p* < 0.01; ****, *p* < 0.0001).

**Figure 7 ijms-24-13436-f007:**
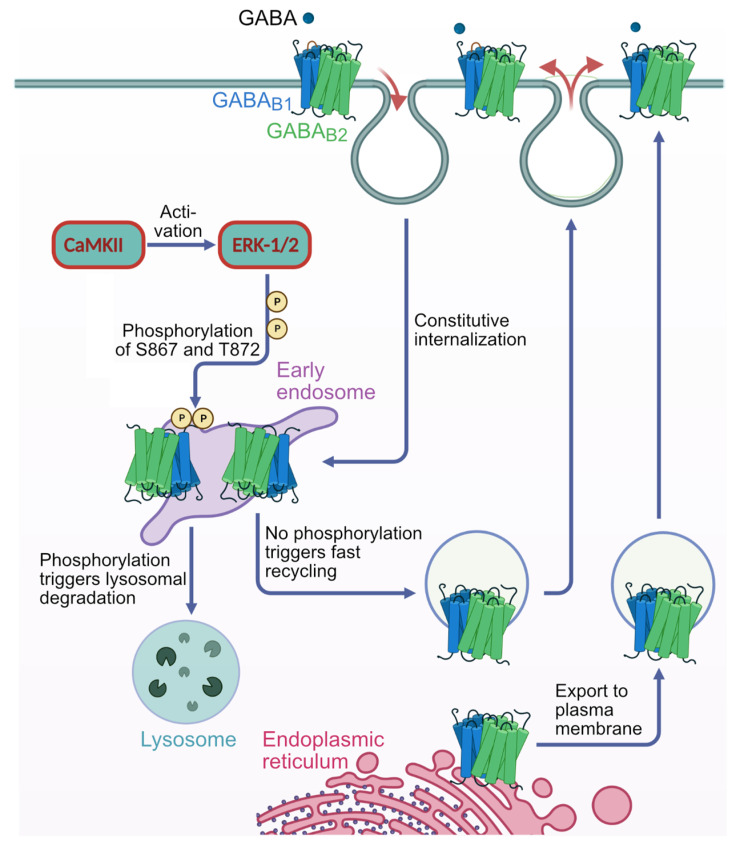
Proposed mechanism of CaMKIIβ and ERK1/2 mediated downregulation/degradation of GABA_B_ receptors. GABA_B_ receptors are constitutively internalized to early endosomes. From early endosomes, receptors are recycled back to the plasma membrane or sorted into lysosomes for degradation. Degraded receptors are replaced by newly synthesized receptors exported from the ER to ensure a constant number of cell surface receptors. The initial signal that tags GABA_B_ receptors for degradation appears to be phosphorylating GABA_B1_ by CaMKIIβ and ERK1/2 at the level of early endosomes. Our data support a mechanism in which CaMKIIβ activates ERK1/2, which phosphorylates GABA_B1_ at S867 and T872 for further sorting of the receptors to lysosomes. As CaMKIIβ most likely does not directly phosphorylate ERK1/2, activated CaMKIIβ might recruit the components of the ERK1/2 activation cascade to GABA_B_ receptors and induce activation of ERK1/2 by phosphorylation of Ref [24,26,28]. This figure was created using BioRender (www.biorender.com, accessed on 12 April 2023).

## Data Availability

The data sets used during the current study are available from the corresponding author upon reasonable request. Raw data have been deposited at ZENODO.org and are publicly available as of the date of publication (DOI: 10.5281/zenodo.8278725).

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
