# Peer review of "ERK1/2-Dependent Phosphorylation of GABAB1(S867/T872), Controlled by CaMKIIβ, Is Required for GABAB Receptor Degradation under Physiological and Pathological Conditions"

_ijms, 2023, doi:10.3390/ijms241713436_

Round 1

Reviewer 1 Report

Using HEK-293 cells and neurons, this manuscript explored the roles of ERK and CaMKIIβ in phosphorylation of GABAB receptors and the potential phosphorylation sites. The overall research idea is clear. But I still have some question.

1. It's easy to understand that HEK-293 cells were transfected with GABAB1 and GABAB2, but for neurons, why did you still transfected neurons with GABAB1 and GABAB2? Have you checked the expression of endogenic  GABAB1 and GABAB2 with over-expression of ERK1/2? 

2. For the treatment duration of Ravoxertinib, I don't think the reference [20] is a good evidence. You should find some papers about Ravoxertinib rather than CaMKII.

3. As you didn't show the results of overexpression of ERK1/2 in neurons in Figure 2, it's insufficient to give conclusion in Line 112-113.

4. For DMSO in Line 135, I think the final concentraion is too high. Usually it's less than 1/1000.

5. In Figure 4, I noticed that the expression of GABABR surface expression in Wt group is so low even through with transfection of wild type HA-tagged GABAB1. Can you give me a explanation?

6. I also found some obvious minor errors in Figures. 1). in Figure 3C, I don't think all the comparisons are necessary. Besides, compared with other figures, the location of "*" needs to be adjusted. 2). in Figure 4A, the title didn't appear completly. Besides, "c'" and "d'" should be "C'" and "D'". 3). in Figure 6A, there's extra A in the "Untreated" group. Besides, in Figure. 6D, the cells in "Ctrl" and "Ctrl+KN93" groups look smaller than the other two groups. Please check and make sure cells in all groups are at the same scale.

7. Also pay attention to the grammar, for example in Line 86, "After 2 days, the cells were tested... ...".

  •  
  •  

There are minor English grammar issues. 

Author Response

  1. It's easy to understand that HEK-293 cells were transfected with GABAB1 and GABAB2, but for neurons, why did you still transfected neurons with GABAB1 and GABAB2? Have you checked the expression of endogenic GABAB1 and GABAB2 with over-expression of ERK1/2?

Response: If possible, we avoided overexpression proteins in neurons to be more closely to physiological conditions. Therefore, we did not over-express ERK1/2 in neurons to test the effect of ERK on GABAB receptors. Instead, we specifically inhibited ERK1/2 activity with Ravoxertinib to test for their effect on the receptors.

Only in experiments with mutant GABAB1 subunits we needed to transfect neurons with GABAB receptor subunits to be able to test their effects. Neurons transfected with wild type GABAB receptor subunits served as corresponding controls in these experiments.

  1. For the treatment duration of Ravoxertinib, I don't think the reference [20] is a good evidence. You should find some papers about Ravoxertinib rather than CaMKII.

Response: The reason of using 10 min of Ravoxertinib incubation might not be sufficiently explained in the manuscript. Inhibition of CaMKII showed its effect within 5-10 min. Since we were interested in the effect of ERK in the pathway triggered by CaMKII, we opted for the same time frame. We revised our statement in line 106 to: ”Since the downregulation of GABAB receptors was prevented by inhibition of CaMKII within a time frame of 5-10 min in this pathway, we accordingly selected an incubation time of 10 min for the inhibition of ERK1/2.”

  1. As you didn't show the results of overexpression of ERK1/2 in neurons in Figure 2, it's insufficient to give conclusion in Line 112-113.

Response: As the reviewer suggested, we removed this conclusion.

  1. For DMSO in Line 135, I think the final concentraion is too high. Usually it's less than 1/1000.

Response: Ravoxertinib was provided by Selleck chemicals as a solution in DMSO. Therefore, we could not reduce the DMSO concentration in our experiments. Because the final concentration of DMSO (1/100) is quite high, we determined the effect of DMSO without Ravoxertinib as a second control. 

  1. In Figure 4, I noticed that the expression of GABABR surface expression in Wt group is so low even through with transfection of wild type HA-tagged GABAB1. Can you give me a explanation?

Response: We had to set the imaging parameters for the cells showing the highest expression level to avoid saturation of the signals. For proper signal quantification, these parameters had to be kept constant for all conditions in an experiment and had to be imaged in one continuous session. This caused weak signals in the wild type group.

  1. I also found some obvious minor errors in Figures.

1). in Figure 3C, I don't think all the comparisons are necessary. Besides, compared with other figures, the location of "*" needs to be adjusted.

Response: The upper continues lines in Fig 3B and C depict the statistical evaluation between the wild type control and all other conditions. Therefore, the stars are correctly positioned. However, this was not made clear in the figure legend and may cause confusion. We now clarified this issue by stating “The upper line depicts the statistical evaluation between the wild type control and all other conditions.” in line 195 and line 206 in the legend to figure 3.

2). in Figure 4A, the title didn't appear completly. Besides, "c'" and "d'" should be "C'" and "D'".

Response: The title line of Fig. 4A is slightly cut off at the top due to the insertion of the figure into the text. This is not the case in the original figure, which will be submitted for assembly of the paper for publishing. We converted "c'" and "d'" to "C'" and "D'".

3). in Figure 6A, there's extra A in the "Untreated" group. Besides, in Figure. 6D, the cells in "Ctrl" and "Ctrl+KN93" groups look smaller than the other two groups. Please check and make sure cells in all groups are at the same scale.

Response: We removed the black extra “A” in Fig. 6A.

All images shown in Fig. 6D are at the same scale. The slightly different sizes of the neurons reflect the heterogeneity of the neurons in the culture and is normal and thus representative for this figure.

  1. Also pay attention to the grammar, for example in Line 86, "After 2 days, the cells were tested... ...".

Response: We checked the manuscript for grammar errors and adjusted it accordingly.

Reviewer 2 Report

The manuscript “ERK1/2-dependent phosphorylation of GABAB1(S867/T872), controlled by CaMKIIbeta, is required for GABAB receptor degradation under physiological and pathological conditions” by Bhat al is a research article which examined whether the phosphorylation of T872 is involved in the downregulation of the GABAB receptors and whether phosphorylation of T872 is mediated by CaMKIIbeta or another protein kinase. The authors found that the mutational inactivation of T872 in GABAB1 suppressed the degradation of GABAB receptors in cultured neurons. Also, the authors found that not only CaMKIIbeta but also ERK1/2 are involved in the degradation pathway of GABAB receptors under both physiological and ischemic conditions. Interestingly, CaMKIIbeta seems not to directly phosphorylate S867, but activates ERK1/2 which then phosphorylate S867 and T872 in GABAB1 receptors. Finally, the authors demonstrated that the blockade of ERK activity after subjecting neurons to ischemic stress prevented the downregulation of GABAB receptor expression. Therefore, the authors suggest that the prevention of ERK1/2-mediated phosphorylation of S867/T872 in GABAB1 receptors suppresses the pathological downregulation of the GABAB receptors after ischemic stress. In general, this study is critical in this field and scientifically sound and contains essential findings. However, I have some issues before publication.

1. The statistical significance was assessed using Student’s t-test and ANOVA. The values of “t” and “F” should be given if possible.

2. Introduction: In this study, the authors focus on the activity of GABAB receptors. However, The balance between excitation and inhibition is mediated by GABAB receptors as well as GABAA receptors. At least, the authors should describe the important role of GABAA receptors in the regulation of neural excitation.

Author Response

  1. The statistical significance was assessed using Student’s t-test and ANOVA. The values of “t” and “F” should be given if possible.

Response: We prefer not to include “t” and “F” values as they would considerably lengthen the respective text and, in our opinion, would compromise the readability of the text. However, if they are considered absolutely essential we can include them.

  1. Introduction: In this study, the authors focus on the activity of GABAB receptors. However, The balance between excitation and inhibition is mediated by GABAB receptors as well as GABAA receptors. At least, the authors should describe the important role of GABAA receptors in the regulation of neural excitation.

Response: The excitation/inhibition balance is affected/regulated by numerous factors with GABA receptors (GABAA and GABAB type receptor) being main players. As our manuscript focuses on the regulation of GABAB receptors, the “Introduction” is streamlined accordingly. We believe that including a discussion on GABAA receptors will render the “Introduction” less concise and do not add essential information required in this study. However, to make it clear that GABAB receptors is not the only factor that regulates the excitation/inhibition balance we changed the sentence in line 34 to: “Under physiological conditions, neuronal excitation is controlled, among other factors, by GABAB receptors.

Round 2

Reviewer 1 Report

After revision, the manuscript becomes much better. 

I noticed that they revised a lot about the English grammar, but I don't think all the revisions are correct. For example, in line 419, "activates" to "activate". Please revise all similar issues before the manuscript is accepted.

  •  

Author Response

Thank you very much for the detailed reading of our manuscript. A native English-speaking colleague checked the manuscript grammatical errors. He found a few, which we accordingly corrected.